# Bringing Graphs to the Table: Zero-Shot Node Classification via Tabular Foundation Models

## Abstract

Graph foundation models (GFMs) have recently emerged as a promising paradigm for achieving broad generalization across various graph data. However, existing GFMs are often trained on datasets that may not fully reflect real-world graphs, limiting their generalization performance. In contrast, tabular foundation models (TFMs) not only excel at classical tabular prediction tasks but have also shown strong applicability in other domains such as time series forecasting, natural language processing, and computer vision. Motivated by this, we take an alternative view to the standard perspective of GFMs and reformulate node classification as a tabular problem. In this reformulation, each node is represented as a row with feature, structure, and label information as columns, enabling TFMs to directly perform *zero-shot* node classification via in-context learning. In this work, we introduce TAG, a tabular approach for graph learning that first converts a graph into a table via feature and structural encoders, applies multiple TFMs to diversely subsampled tables, and then aggregates their outputs through ensemble selection. Through experiments on 28 real-world datasets, TAG achieves consistent improvements over task-specific GNNs and state-of-the-art GFMs, highlighting the potential of tabular reformulation for scalable and generalizable graph learning.

## 1 Introduction

Graph foundation models (GFMs) have recently emerged as a central research direction in graph machine learning (Mao et al., 2024; Huang et al., 2025). However, their effectiveness on node classification benchmarks has so far been limited, with performance improvements over standard task-specific graph neural networks (GNNs) (Kipf & Welling, 2017; Veličković et al., 2018) often being marginal (Zhao et al., 2025; Finkelshtein et al., 2025). A growing body of work argues that this shortfall is primarily due to the characteristic of the training data: many available pretraining graphs are small in scale, contain outdated node features, and rely on heuristic or artificial topological structures, making them poor representatives of real-world graphs (Bechler-Speicher et al., 2025; Platonov et al., 2023a; Coupette et al., 2025).

In contrast, tabular foundation models (TFMs) have demonstrated broad applicability by representing heterogeneous domains in a unified tabular format. This perspective has enabled strong generalization in settings as diverse as computer vision (McCarter, 2025), natural language processing (Van Breugel & Van Der Schaar, 2024; Hegselmann et al., 2023; Mráz et al., 2025), and time-series forecasting (Hoo et al., 2025). This motivates our central research question: *Can the generalization strengths of tabular foundation models be effectively leveraged to build a foundation model for node classification?*

To address this question, we first observe that once the graph's topological structure has been exploited, either through neighbor aggregation or through least-squares solutions in GraphAny (Zhao et al., 2025) and TS-GNNs (Finkelshtein et al., 2025), the node classification problem reduces to predicting labels from feature vectors. At this stage, the goal is

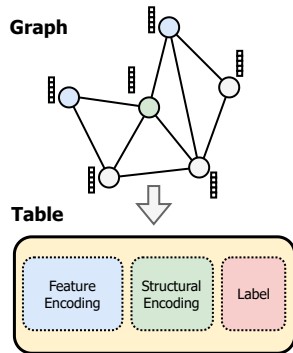

**Graph**

**Table**

Feature Encoding | Structural Encoding | Label

Figure 1: An illustration of how TAG transforms a graph into a tabular representation using feature and structural encoders.

simply to map vectors to labels. In traditional GNNs and TS-GNNs, this is typically done with a lightweight classifier, whereas GraphAny employs a more elaborate attention mechanism to produce the final predictions. This perspective naturally aligns node classification with tabular learning, allowing us to leverage TFMs.

Motivated by this perspective, we introduce TAG, a tabular approach for graph learning that reinterprets node classification as an ensemble learning problem over tabular models. TAG first converts the input graph to a table by computing node-level features using pre-defined feature and structural encoders, as shown in Figure 1. However, the resulting tables are often too large and contain diverse features and label spaces that current TFMs (Hollmann et al., 2023) are not designed to handle. We employ an ensemble aggregation strategy: subsampling multiple smaller, size-constrained tables, applying TFM to each table to obtain individual predictions, and aggregating them via ensemble selection (Caruana et al., 2004). The resulting design unifies graph-specific insights with advances in tabular learning, yielding a single, generalizable foundation model that does not require pretraining on graph data, yet still outperforms the state-of-the-art GFMs and task-specific GNNs by 7%.

**Contributions.** Our work makes the following contributions toward building generalizable foundation models for graph tasks:

1. We formulate *node classification* as a *tabular classification* problem, which enables the use of *tabular foundation models* trained exclusively on tabular data for *zero-shot inference*, without restrictions on the number of features, labeled nodes, or output classes.

2. We introduce TAG, a tabular approach for graph learning that represents nodes as rows in a table and adapts TFMs to node classification via subsampling and ensemble aggregation.

3. We evaluate our approach on 28 real-world node classification datasets, demonstrating significant improvements over existing GFMs and task-specific GNNs, improving the averaged accuracy from 65.78% to 73.10%.

## 2 RELATED WORK

**Graph foundation models for node classification.** Graph Neural Networks (GNNs) are the dominant approach for solving graph machine learning tasks. However, these models are typically trained separately on each dataset and lack the ability to generalize across different feature and label spaces. Graph Foundation Models (GFMs) have emerged (Mao et al., 2024; Huang et al., 2025) to address this gap by learning transferable representations from diverse graph data. GFMs have shown strong performance on the *label inpainting* problem (Finkelshtein et al., 2025), a subtask of inductive node-based learning where the test graph contains partially observed features, analogous to image inpainting. One of the first attempts in this direction is GraphAny (Zhao et al., 2025), which characterizes the natural symmetries required of a GFM: node permutation-equivariance, label permutation-equivariance, and feature permutation-invariance. It then constructs an ensemble by combining the closed-form solutions of multiple least-squares models that respect these symmetries. An attention mechanism is used to weight the ensemble components, and the resulting model has been shown to generalize to unseen graphs with arbitrary feature and label spaces. The subsequent TS-GNN model (Finkelshtein et al., 2025) formalizes the aforementioned symmetries into a theoretically grounded framework for GFMs. It derives linear layers that respect the same symmetry constraints, proves the universality of the resulting architecture over multi-sets, and shows empirically that performance improves as the number of training graphs increases. Our proposed method TAG, incorporates tabular foundation models to allow for a richer set of ensemble models, leading to significantly improved generalization capabilities and empirical performance.

**Tabular foundation models and their application.** A tabular learning problem involves predicting labels from data organized in a table, where each row corresponds to an instance and columns correspond to features. Unlike structured domains such as images or text, tabular data are heterogeneous and lack strong inductive biases, which historically limit transferability across datasets.

Prior-Data Fitted Networks (PFNs) (Müller et al., 2022) are transformer-based models that enable in-context learning (ICL) on tabular datasets. The distinctive ingredient in PFNs is their *synthetic-task pretraining*: millions of tabular datasets are sampled from randomized structural causal models spanning diverse mechanisms (linear, tree-based, interaction-heavy, etc.). Each synthetic dataset

is split into a context and a query set, and the transformer is trained to approximate the posterior predictive distribution on the queries under the known generative process. This curriculum induces a broad prior over tabular problems, enabling state-of-the-art tabular foundation models (TFM) such as TabPFN (Hollmann et al., 2023; 2025) to adapt to *new* tables without gradient updates.

Beyond classical tabular benchmarks, TFMs have shown surprising generalization capabilities in other domains, including computer vision (McCarter, 2025), natural language processing (Van Breugel & Van Der Schaar, 2024; Hegselmann et al., 2023; Mráz et al., 2025), and time-series forecasting (Hoo et al., 2025). These results suggest that TFMs are not limited to standard tabular tasks, but can serve as a unifying framework for heterogeneous data. Our proposed model, TAG, builds on this insight by framing node classification as a tabular learning problem, enabling TFMs to adapt to new graphs without retraining and bridging their advances with the challenges of graph machine learning.

Concurrent to our work, Eremeev et al. (2025) propose G2T-FM, which augments node features with structural signals, converts them into tabular rows, and applies a TFM, conceptually similar to TAG. G2T-FM inherits TabPFN's limits and is thus restricted to graphs with $\leq 10{,}000$ labeled nodes, $\leq 500$ node features, and $\leq 10$ classes. In contrast, TAG's table subsampling and ensemble selection avoid these caps, supporting graphs with an arbitrary number of classes, labeled nodes, and node features.

## 3 BACKGROUND: GRAPHS, LABEL INPAINTING AND TABPFN

**Graphs.** We consider a simple, undirected[1], unweighted graph $G = (V, E, \boldsymbol{X}, \boldsymbol{Y})$ with $N$ nodes. The graph structure $(V, E)$ is represented by an adjacency matrix $\boldsymbol{A} \in \{0, 1\}^{N \times N}$. Each node is equipped with a feature vector and a class label: the feature vectors are collected in the matrix $\boldsymbol{X} \in \mathbb{R}^{N \times F}$, and the one-hot encoded labels across $C$ classes are given by $\boldsymbol{Y} \in \{0, 1\}^{N \times C}$. The (random-walk) *normalized adjacency matrix* is defined as $\hat{\boldsymbol{A}} = \boldsymbol{D}^{-1} \boldsymbol{A}$, where $\boldsymbol{D}$ is a diagonal degree matrix $\boldsymbol{D} = \text{diag}(d_1, \ldots, d_n)$ with $d_i = \sum_{j=1}^{n} \boldsymbol{A}_{ij}$ denoting the degree of node $i$. Given a matrix $\boldsymbol{M} \in \mathbb{R}^{N \times D}$ and a subset of nodes $S \subseteq V$, we write $\boldsymbol{M}_S \in \mathbb{R}^{|S| \times D}$ for the submatrix consisting of the rows of $\boldsymbol{M}$ indexed by $S$. For a single node $v \in V$, the corresponding row vector is denoted by $\boldsymbol{m}_v \in \mathbb{R}^D$. Lastly, we denote $[N] = \{1, 2, \ldots, N\}$.

**Label Inpainting.** We study the label inpainting (Finkelshtein et al., 2025) setting, a subtask of inductive node classification. Let $L \subset V$ denote the set of *labeled nodes* with labels $\boldsymbol{Y}_L \in \mathbb{R}^{|L| \times C}$, and let $Q \subseteq V \setminus L$ denote the set of *query nodes*. The goal is to predict the missing labels $\boldsymbol{Y}_Q \in \mathbb{R}^{|Q| \times C}$ for the query nodes, given the existing labels. Unlike the classical semi-supervised regime, which assumes a fixed set of training labels, label inpainting treats the labeled nodes themselves as part of the input, and the test graphs are also partially labeled. This formulation enables generalization across varying label sets and unseen graphs: given a partially labeled graph, the task is to "fill in" the missing labels by leveraging both node features and structural information, analogous to inpainting in computer vision.

**TabPFN.** A state-of-the-art tabular foundation model that performs in-context learning by amortizing Bayesian inference (Müller et al., 2022). Given a labeled context $(\boldsymbol{X}_L, \boldsymbol{Y}_L)$ with $\boldsymbol{X}_L \in \mathbb{R}^{|L| \times F}$, $\boldsymbol{Y}_L \in \{0, 1\}^{|L| \times C}$, and a query set $\boldsymbol{X}_Q \in \mathbb{R}^{|Q| \times F}$, TabPFN outputs class probabilities

$$\hat{\boldsymbol{Y}}_Q = \text{TabPFN}\big((\boldsymbol{X}_L, \boldsymbol{Y}_L), \boldsymbol{X}_Q\big) \in [0, 1]^{|Q| \times C},$$

with each row summing to one. For each $q \in Q$, the prediction $\hat{\boldsymbol{y}}_q$ approximates the posterior predictive distribution $p(\mathbf{y}_q \mid \mathbf{x}_q, \boldsymbol{X}_L, \boldsymbol{Y}_L)$, under a prior learned from large-scale synthetic data-generating processes during pretraining. In our node-classification setting, each row corresponds to a node.

## 4 THE TAG FRAMEWORK

In this section, we present TAG, a tabular approach to graph learning that reinterprets node classification as an ensemble learning problem over tabular models. Given a graph $G = (V, E, \boldsymbol{X}, \boldsymbol{Y})$, our key idea is to reduce graph-structured data into a tabular form that can be directly processed by powerful tabular foundation models. TAG decouples the problem into three stages:

---

[1]All results naturally extend to directed graphs; we focus on undirected graphs for ease of presentation.

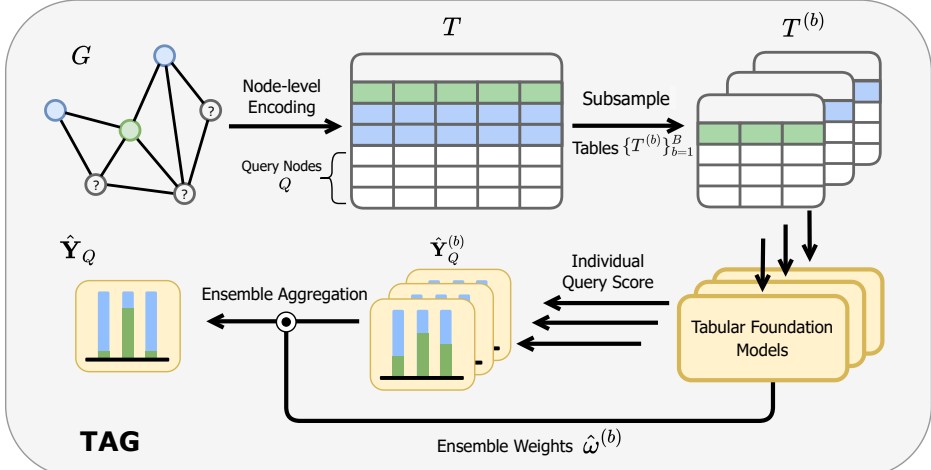

Figure 2: Overall pipeline of TAG. Given a graph $G$ and a set of querying nodes $Q$, TAG first employs node-level encoding on $G$ and converts it into a table $T$. Then TAG constructs $\{b\}_{b=1}^{B}$ subsampled tables and applies TFM on them. Finally, TAG aggregates individual query scores $\hat{Y}_Q^{(b)}$ via ensemble selection to produce the final prediction $\hat{Y}_Q$.

1. **Node-level encoding:** Each node is represented as a row in a table by concatenating its label (if available) with multiple feature and structure encoders. This step transforms graph information into a tabular format that can be processed by tabular models.

2. **Tabular learning:** Since TFMs are limited to relatively small tables, we construct multiple subsampled tables by selecting subsets of columns and labeled rows, and obtain separate predictions from the TFM given each subsample.

3. **Ensemble aggregation:** Predictions from the subsampled tables are combined through ensemble selection, which uses predictions on held-out data to determine ensemble weights that both account for model quality and model interactions. The final prediction for each query node is obtained as a weighted combination of these outputs.

This modular design allows TAG to exploit the strengths of tabular foundation models while remaining lightweight and training-free. Figure 2 presents the overall pipeline of TAG.

## 4.1 NODE-LEVEL ENCODING

Given $G = (V, E, \boldsymbol{X}, \boldsymbol{Y})$, we employ $I+1$ feature encoders $\{\boldsymbol{\phi}^{(i)}\}_{i=0}^{I} : \mathbb{R}^{N \times (F+N)} \to \mathbb{R}^{N \times D_1}$ and $J$ structure encoders $\{\boldsymbol{\psi}^{(j)}\}_{j=1}^{J} : \mathbb{R}^{N \times N} \to \mathbb{R}^{N \times D_2}$. These produce a tabular representation $\boldsymbol{T} \in \mathbb{R}^{N \times (ID_1 + JD_2 + C)}$, where each node $v \in V$ corresponds to a row defined as

$$\boldsymbol{T}_v = \left( \boldsymbol{\phi}_v^{(0)}(\boldsymbol{X}, \boldsymbol{A}), \dots, \boldsymbol{\phi}_v^{(I)}(\boldsymbol{X}, \boldsymbol{A}), \boldsymbol{\psi}_v^{(1)}(\boldsymbol{A}), \dots, \boldsymbol{\psi}_v^{(J)}(\boldsymbol{A}), \boldsymbol{Y}_v \right),$$

where $\boldsymbol{\phi}_v^{(i)}(\boldsymbol{X}, \boldsymbol{A}) \in \mathbb{R}^{D_1}$ and $\boldsymbol{\psi}_v^{(j)}(\boldsymbol{A}) \in \mathbb{R}^{D_2}$ denote the outputs of the $i$-th feature encoder and the $j$-th structure encoder for node $v$, respectively. The term $\boldsymbol{Y}_v$ is the one-hot label encoding if $v \in L$, and the zero-vector otherwise.

**Feature encoders.** We include the raw node features $\boldsymbol{\phi}^{(0)}(\boldsymbol{X}, \boldsymbol{A}) = \boldsymbol{X}$. To incorporate local structural information, we add neighborhood-smoothed features, which have been shown to improve performance (see Section 5.3). Specifically, we use $k$-order neighborhood averages for $k \in \{1, \cdots, 4\}$, i.e., $\boldsymbol{\phi}^{(k)}(\boldsymbol{X}, \boldsymbol{A}) = \hat{\boldsymbol{A}}^k \boldsymbol{X}$, where $\hat{\boldsymbol{A}}$ is the normalized adjacency matrix.

**Structure encoders.** Unlike GNNs, TFMs operate on sets of rows and are thus unaware of the underlying topology. To reintroduce this information, we employ well-established structural encoders designed to capture both local and global graph structure. Specifically, we include (1) RandomWalkPE (Dwivedi et al., 2022), denoted as $\boldsymbol{\psi}^{(1)}$, which encodes local structures, (2)

LaplacianEigenvectorPE (Dwivedi et al., 2023), denoted as $\psi^{(2)}$, the top-20 eigenvectors of the normalized Laplacian providing smooth global encodings, and (3) GPSE (Cantürk et al., 2024), denoted as $\psi^{(3)}$, pretrained embeddings from frozen message-passing networks that have been shown to be strong general-purpose structure encoders.

**Remark 4.1.** *The construction of both feature and structure encodings is* lightweight and training-free, *relying only on closed-form computations.*

### 4.2 TABULAR LEARNING AND ENSEMBLE AGGREGATION

**Tabular learning.** The tabular representation of a graph is denoted by $\boldsymbol{T} = (\boldsymbol{Z}, \boldsymbol{Y})$, where $\boldsymbol{Z} \in \mathbb{R}^{N \times (ID_1 + JD_2)}$ stores the *row features* obtained by concatenating all feature and structure encodings, and $\boldsymbol{Y} \in \{0,1\}^{N \times C}$ contains the corresponding row one-hot label vectors. We distinguish between *labeled rows* $(\boldsymbol{Z}_L, \boldsymbol{Y}_L) \in \mathbb{R}^{|L| \times (ID_1 + JD_2 + C)}$, which include both row features and labels, and *query rows* $\boldsymbol{Z}_Q \in \mathbb{R}^{|Q| \times (ID_1 + JD_2)}$, which include only row features. The task is to inpaint the missing labels $\hat{\boldsymbol{Y}}_Q$ for the query rows, which could be achieved by directly applying TabPFN as

$$\hat{\boldsymbol{Y}}_Q \;=\; \mathrm{TabPFN}\big((\boldsymbol{Z}_L, \boldsymbol{Y}_L),\, \boldsymbol{Z}_Q\big). \tag{1}$$

However, due to the computational complexity of its attention mechanism and its pretraining regime, TabPFN is currently limited to relatively small tables, supporting at most 10,000 labeled rows, 500 columns, and 10 classes. This restriction prevents it from serving as a general-purpose foundation model, since the encoded tables in TAG can reach up to 90,000 labeled rows, 40,000 columns, and 70 different classes, exceeding the capability of TabPFN.

**Subsampling.** We address this limitation by constructing $B$ smaller subsampled tables $\{\boldsymbol{T}^{(b)}\}_{b=1}^B$, each fitting TabPFN's size constraints. Each $\boldsymbol{T}^{(b)}$ is obtained by (1) retaining all unlabeled rows, (2) uniformly subsampling columns, and (3) class-balanced subsampling of labeled rows to preserve its distribution. TabPFN is then applied to each $\boldsymbol{T}^{(b)}$, producing predictions for the query rows $\hat{\boldsymbol{Y}}_Q^{(b)}$. For tables with more than 10 classes, we adopt an error-correcting output code (ECOC) strategy (Dietterich & Bakiri, 1994) suggested by Hollmann et al. (2025), which splits tasks with more than ten classes into subproblems with 10 or fewer classes.

**Ensemble aggregation.** We aggregate the predictions $\{\hat{\boldsymbol{Y}}_Q^{(b)}\}_{b=1}^B$ through an affine combination, where the contribution of each predictor is determined via *ensemble selection (ES)* (Caruana et al., 2004). A *hold-out set* $H \subset L$, is randomly sampled from the labeled nodes, with the remaining nodes $A = L \setminus H$ called *anchor nodes*. Each TabPFN model applies subsampling independently to generate its own table $\tilde{\boldsymbol{T}}^{(b)}$ from the context $(\boldsymbol{X}_A, \boldsymbol{Y}_A)$, where the subset of columns sampled per model stays fixed between ensemble selection and inference. We then create held-out predictions per model

$$\hat{\boldsymbol{Y}}_H^{(b)} \;=\; \mathrm{TabPFN}\big(\tilde{\boldsymbol{T}}^{(b)}, \boldsymbol{Z}_H^{(b)}\big),$$

where $\boldsymbol{Z}_H^{(b)}$ denotes the column-subsampled features of the held-out rows. Ensemble selection (Caruana et al., 2004) uses greedy forward selection to approximate weights $\hat{w}^{(b)}$ for the intractable problem

$$(\hat{w}^{(1)}, \ldots, \hat{w}^{(B)}) = \underset{\substack{\forall b \in [B],\, w^{(b)} \in \mathbb{R} \\ \sum_{b=1}^B w^{(b)} = 1}}{\arg\max} \; \mathrm{Acc}_H\Big(\sum_{b=1}^B w^{(b)} \hat{\boldsymbol{Y}}_H^{(b)}, \boldsymbol{Y}_H\Big),$$

where $\mathrm{Acc}_H$ is the standard multi-class accuracy over the held-out set $H$

$$\mathrm{Acc}_H(\hat{\boldsymbol{Y}}, \boldsymbol{Y}) = \frac{1}{|H|} \sum_{i \in H} \mathbb{1}\Big\{ \underset{c \in [C]}{\arg\max}\, \hat{\boldsymbol{Y}}_{i,c} \;=\; \underset{c \in [C]}{\arg\max}\, \boldsymbol{Y}_{i,c} \Big\},$$

and $\mathbb{1}\{\cdot\}$ is the indicator function (1 if the condition holds, 0 otherwise). The optimized weights are then used to combine the query predictions, yielding the final ensemble prediction as

$$\hat{\boldsymbol{Y}}_Q = \sum_{b=1}^B \hat{w}^{(b)} \hat{\boldsymbol{Y}}_Q^{(b)}.$$

**Remark 4.2.** *While we focus on TabPFN for its zero-shot in-context prediction, our framework is general and can accommodate alternative tabular learners (e.g., gradient-boosted trees).*

**LinearGNNs.** In parallel to TabPFN, we train LinearGNNs (Zhao et al., 2025) on each encoded matrix $\boldsymbol{\eta} \in \{\boldsymbol{\phi}^{(i)}(\boldsymbol{X}, \boldsymbol{A})\}_{i=0}^{4} \cup \{\boldsymbol{\psi}^{(j)}(\boldsymbol{A})\}_{j=1}^{3}$, yielding eight additional lightweight ensemble members. For an encoder $\boldsymbol{\eta} \in \mathbb{R}^{N \times D}$, we fit a separate linear map $\boldsymbol{W} \in \mathbb{R}^{D \times C}$ on the labeled nodes $L$ via the close form solution of the least squares problem

$$\boldsymbol{W}^{\star} = \underset{\boldsymbol{W} \in \mathbb{R}^{F \times C}}{\arg \min} \left\| \boldsymbol{\eta} \boldsymbol{W} - \boldsymbol{Y}_L \right\|_F^2.$$

Predictions are then obtained for all nodes as class logits $\hat{\boldsymbol{Y}} = \boldsymbol{\eta} \boldsymbol{W}^{\star}$, yielding a training-free, closed-form predictor for each of our encoders.

## 5 EXPERIMENTS

In this section, we conduct experiments addressing the following research questions:

**Q1** How does TAG's zero-shot generalization compare to GFMs and end-to-end GNNs?

**Q2** Can tabular foundation models be fine-tuned for node classification?

Furthermore, we ablate each key component of TAG to quantify its contribution: feature encoders (Section 5.3), structure encoders (Appendix A.1), the number of subsampled tables (Section 5.4), ensemble selection, and LinearGNN (Section 5.5).

**Datasets.** Following Finkelshtein et al. (2025), we evaluate on 28 diverse node-classification datasets using their official splits. These include *brazil*, *usa*, and *europe* (Ribeiro et al., 2017); *chameleon*, *squirrel* (Rozemberczki et al., 2021); *roman-empire*, *amazon-ratings*, *minesweeper*, *questions*, and *tolokers* (Platonov et al., 2023b). We further employ *wiki-attr* and *blogcatalog* (Yang et al., 2023); *cornell*, *wisconsin*, *texas*, and *actor* (Pei et al., 2020); and the classical benchmarks *cora*, *citeseer*, and *pubmed* (Yang et al., 2016). In addition, we consider *co-cs*, *co-physics*, *computers*, and *photo* (Shchur et al., 2023); *full-dblp* and *full-cora* (Bojchevski & Günnemann, 2018); *wiki-cs* (Mernyei & Cangea, 2022); and *last-fm-asia* and *deezer* (Rozemberczki & Sarkar, 2020). Finally, we include the large-scale *arxiv* dataset (Hu et al., 2021). Dataset statistics can be found in Appendix B.

All experiments are conducted on a single NVIDIA RTX PRO 6000. Our codebase is publicly available at: `https://anonymous.4open.science/r/tag_iclr_codebase-5971/`. All results reflect the mean accuracy and standard deviation over 5 random seeds.

### 5.1 COMPARISON OF TAG WITH GNNS AND GFMS

We evaluate whether TAG can leverage a pretrained TFM to deliver strong zero-shot performance on node classification across diverse domains, feature spaces, graph topologies, and label sets (**Q1**).

**Baselines.** We compare TAG with end-to-end MeanGNN (Finkelshtein et al., 2024) and GAT (Veličković et al., 2018), trained separately per dataset. We also compare with GFMs GraphAny (Zhao et al., 2025) and TS-Mean (Finkelshtein et al., 2025), pretrained on the *cora* dataset. We apply PCA to 2048 components before training on *full-Cora*, *co-cs*, and *co-physics* following the setup in Finkelshtein et al. (2025). TAG uses a pretrained TFM (TabPFN) without extra pretraining or fine-tuning. For inference, we ensemble predictions from 10 subsampled tables and 8 LinearGNN variants.

**Key methodological differences between baselines.** The three approaches GNNs, GFMs and TAG differ along three axes. *(i) How structure is used:* end-to-end GNNs and GFMs model graphs explicitly via message passing, whereas TAG employs a structure-agnostic transformer (TabPFN) and injects graph information through feature/structure encoders and LinearGNN-derived signals. *(ii) Pretraining scale:* end-to-end GNNs lack cross-dataset pretraining and current GFMs see only a small number of real graphs, while TAG inherits a broad prior from TabPFN trained on a massive corpus of tabular tasks. *(iii) Data realism:* GNNs and GFMs are typically trained on real-world graph datasets, whereas TabPFN is trained solely on synthetic data—trading realism for scale and diversity. Our experiments test whether this synthetic-but-vast prior transfers effectively to graphs when coupled with a suitable node-level encoding strategy.

Table 1: Accuracy on 28 node classification datasets. End-to-end (MeanGNN, GAT) are trained per dataset; GFMs (GraphAny, TS-Mean) are pretrained on *cora*; TAG uses pretrained TabPFN models.

| Dataset | End-to-end | | GFM | | TAG |
|---|---|---|---|---|---|
| | MeanGNN | GAT | GraphAny | TS-Mean | |
| actor | 32.03±0.29 | 32.59±0.83 | 27.54±0.20 | 28.09±0.93 | 31.18±0.18 |
| amazon-ratings | 40.74±0.13 | 40.63±0.66 | 42.80±0.09 | 42.27±1.40 | 44.34±0.32 |
| arxiv | 50.94±0.38 | 57.93±3.44 | 58.85±0.03 | 56.33±2.58 | 66.70±0.21 |
| blogcatalog | 84.48±0.74 | 78.20±7.23 | 71.54±3.04 | 76.30±2.92 | 79.77±1.06 |
| brazil | 32.31±7.50 | 35.38±4.21 | 33.84±15.65 | 39.23±5.70 | 73.08±4.03 |
| chameleon | 61.75±0.94 | 58.46±6.23 | 63.64±1.48 | 60.83±5.41 | 72.94±1.13 |
| citeseer | 65.06±1.30 | 63.92±0.84 | 68.88±0.10 | 68.66±0.19 | 68.08±0.61 |
| co-cs | 80.87±0.69 | 82.28±0.86 | 90.06±0.80 | 90.92±0.47 | 90.49±0.50 |
| co-physics | 79.05±1.13 | 85.92±1.10 | 91.85±0.34 | 92.61±0.61 | 92.31±0.27 |
| computers | 73.88±0.88 | 70.94±3.40 | 82.79±1.13 | 81.37±1.25 | 84.33±0.33 |
| cornell | 63.78±1.48 | 69.73±2.26 | 63.24±1.32 | 68.65±2.42 | 75.14±2.16 |
| deezer | 54.91±1.81 | 55.22±2.33 | 51.82±2.49 | 52.31±2.52 | 52.99±1.65 |
| europe | 39.12±6.44 | 39.00±4.30 | 41.25±7.25 | 35.88±6.91 | 55.38±2.30 |
| full-DBLP | 65.01±2.40 | 67.34±2.75 | 71.48±1.44 | 66.42±3.65 | 71.92±1.46 |
| full-cora | 55.85±1.04 | 59.95±0.88 | 51.18±0.78 | 53.58±0.73 | 54.56±0.19 |
| last-fm-asia | 77.23±1.01 | 72.65±0.48 | 81.14±0.42 | 78.03±1.02 | 85.47±0.29 |
| minesweeper | 84.06±0.18 | 84.15±0.24 | 80.46±0.11 | 80.68±0.38 | 85.83±0.13 |
| photo | 88.95±1.08 | 80.78±3.59 | 89.91±0.88 | 90.18±1.30 | 89.63±0.48 |
| pubmed | 74.56±0.13 | 75.12±0.89 | 76.46±0.08 | 74.98±0.56 | 78.96±0.43 |
| questions | 97.16±0.06 | 97.13±0.05 | 97.07±0.03 | 97.02±0.01 | 97.14±0.01 |
| roman-empire | 69.37±0.66 | 69.80±4.18 | 63.34±0.58 | 66.36±1.02 | 74.12±0.28 |
| squirrel | 43.32±0.66 | 38.16±1.04 | 49.74±0.47 | 41.81±0.80 | 65.71±0.13 |
| texas | 76.76±1.48 | 81.62±6.45 | 71.35±2.16 | 73.51±4.01 | 81.08±2.09 |
| tolokers | 78.59±0.66 | 78.22±0.37 | 78.20±0.03 | 78.12±0.09 | 82.82±0.15 |
| usa | 43.93±1.16 | 43.03±2.08 | 43.35±1.62 | 42.34±2.12 | 60.50±0.80 |
| wiki-attr | 74.23±0.89 | 68.91±9.50 | 60.27±3.06 | 69.89±1.31 | 70.09±0.92 |
| wiki-cs | 71.97±1.70 | 74.99±0.59 | 74.11±0.60 | 74.16±2.07 | 79.07±0.54 |
| wisconsin | 74.12±12.20 | 73.33±8.27 | 59.61±5.77 | 61.18±11.38 | 83.14±1.33 |
| Average (28 graphs) | 65.50±1.75 | 65.55±2.82 | 65.56±1.86 | 65.78±2.28 | **73.10**±0.15 |

**Results.** TAG displays strong performance across all datasets, either matches or surpasses MeanGNN, GAT, GraphAny and TS-Mean, leading to **an over 7% accuracy increase over state-of-the-art performance.** This establishes TAG as, to our knowledge, the first foundation model with substantial improvements over end-to-end training across a broad suite of graphs (**Q1**).

We observe the largest gains on two families of datasets. *(i) Small-scale geospatial "airline" graphs*, comprising *brazil*, *europe*, and *usa*, where TAG achieves an average improvement of 22% over the best competing GFM. This is likely because in these datasets, the signal is primarily structural, and node features are minimally informative (see Section 5.3). As a result, existing GFM and GNN, which overly rely on message-passing without explicit structure encodings, struggle to capture the relevant topology, whereas TAG's structure encodings make the signal directly accessible. *(ii) Small-scale heterophilic benchmarks*, namely *wisconsin*, *cornell*, and *texas*, where TAG improves by 12% on average. In heterophilous regimes, neighbors tend to have different classes, and the message-passing mechanisms used by GraphAny and TS-Mean can propagate misleading label information. TAG instead leverages a strong tabular prior and variance reduction via ensembling over subsampled tables, which yields stable zero-shot transfer. Although the most pronounced improvements occur on small graphs, TAG also improves performance on larger graphs such as *arxiv*, albeit with smaller margins.

We believe that the overall significant difference in performance can be attributed to the much larger scale of training data available for training the tabular foundation models at the core of TAG. For instance, TabPFN was trained on 130 million synthetic datasets, which allows it to experience a larger data distribution. In contrast, the GFM pretrained on the largest number of datasets to date is a variant of TS-Mean, which was trained only on nine datasets. While training on additional datasets does increase its performance (Finkelshtein et al., 2025), TS-Mean's average accuracy of 68.57% remains substantially below TAG's mean accuracy of 74.39% on the remaining 20 datasets. We

therefore view TAG's advantage as evidence that larger and more diverse pretraining distributions, whether synthetic or real, can substantially improve downstream accuracy.

## 5.2 Fine-tuning tabular foundation models for node classification

Table 2: Accuracy of pretrained vs. fine-tuned TAG on 20 unseen datasets.

| Dataset | Pretrained | Fine-tuned |
|---|---|---|
| arxiv | 66.63±0.57 | 67.06±0.56 |
| full-cora | 45.20±2.62 | 51.23±1.48 |
| citeseer | 59.80±2.30 | 60.02±0.68 |
| full-DBLP | 69.81±4.20 | 69.79±2.78 |
| pubmed | 78.20±1.20 | 78.70±0.19 |
| wiki-attr | 67.88±1.43 | 68.94±1.57 |
| wiki-cs | 78.60±1.13 | 79.31±0.69 |
| blogcatalog | 67.87±2.23 | 70.22±2.66 |
| last-fm-asia | 85.56±0.62 | 85.75±0.45 |
| deezer | 52.71±3.64 | 52.94±3.12 |
| co-cs | 86.46±0.39 | 86.66±0.55 |
| co-physics | 90.00±0.97 | 90.24±1.49 |
| cornell | 69.19±3.08 | 68.11±2.96 |
| wisconsin | 74.51±6.04 | 74.12±8.48 |
| brazil | 76.92±7.69 | 76.15±7.40 |
| chameleon | 71.84±2.41 | 72.46±3.14 |
| squirrel | 66.38±0.63 | 66.21±0.40 |
| amazon-ratings | 44.21±0.55 | 44.30±0.28 |
| minesweeper | 85.61±0.22 | 85.62±0.43 |
| questions | 97.04±0.04 | 97.02±0.00 |
| Average | 71.72±0.74 | 72.24±0.78 |

| | structure | +0-hop | +1-hop | +2-hop | +3-hop | +4-hop |
|---|---|---|---|---|---|---|
| actor | 25.25 | 34.18 | 32.80 | 31.46 | 31.09 | 31.18 |
| amazon-ratings | 43.20 | 42.27 | 43.36 | 43.85 | 43.84 | 44.34 |
| arxiv | 56.00 | 55.45 | 61.98 | 65.59 | 65.88 | 66.70 |
| blogcatalog | 64.69 | 78.99 | 78.68 | 78.60 | 79.12 | 79.77 |
| brazil | 70.77 | 75.38 | 74.62 | 71.54 | 73.08 | 73.08 |
| chameleon | 69.87 | 69.96 | 72.11 | 72.85 | 73.07 | 72.94 |
| citeseer | 43.08 | 61.10 | 65.28 | 68.20 | 65.82 | 68.08 |
| co-cs | 80.48 | 88.57 | 90.43 | 90.89 | 90.71 | 90.97 |
| co-physics | 89.15 | 90.32 | 91.92 | 91.52 | 92.26 | 92.33 |
| computers | 72.02 | 76.12 | 83.03 | 84.54 | 84.53 | 84.33 |
| cornell | 45.41 | 79.46 | 75.68 | 77.84 | 74.05 | 75.14 |
| deezer | 51.82 | 55.33 | 53.80 | 53.23 | 52.49 | 52.99 |
| europe | 52.50 | 53.38 | 51.50 | 57.25 | 56.75 | 55.38 |
| full-DBLP | 57.55 | 60.59 | 65.72 | 72.41 | 71.96 | 71.92 |
| full-cora | 9.45 | 38.46 | 55.07 | 57.37 | 57.94 | 58.45 |
| last-fm-asia | 83.57 | 84.41 | 84.27 | 85.39 | 85.05 | 85.47 |
| minesweeper | 80.00 | 80.00 | 83.74 | 85.26 | 85.61 | 85.83 |
| photo | 82.20 | 87.41 | 90.06 | 90.23 | 90.26 | 89.63 |
| pubmed | 71.22 | 76.46 | 76.50 | 76.38 | 79.18 | 78.96 |
| questions | 97.02 | 97.18 | 97.12 | 97.15 | 97.14 | 97.14 |
| roman-empire | 32.87 | 70.58 | 75.34 | 74.88 | 74.34 | 74.12 |
| squirrel | 61.10 | 55.43 | 63.23 | 65.57 | 65.69 | 65.71 |
| texas | 68.11 | 80.54 | 82.70 | 81.62 | 81.08 | 81.08 |
| tolokers | 80.91 | 83.03 | 82.76 | 83.19 | 82.90 | 82.82 |
| usa | 59.10 | 58.67 | 59.89 | 61.05 | 60.54 | 60.50 |
| wiki-attr | 59.92 | 71.71 | 70.59 | 71.15 | 70.51 | 70.09 |
| wiki-cs | 74.88 | 78.07 | 78.29 | 78.97 | 79.23 | 79.07 |
| wisconsin | 58.43 | 82.75 | 85.49 | 85.49 | 83.53 | 83.14 |
| Average | 62.16 | 70.21 | 72.36 | 73.34 | 73.13 | 73.26 |

Figure 3: Mean accuracy of TAG with varying feature encoder depth.

Zero-shot transfer is attractive for out-of-the-box deployment, but practitioners often have a few labeled graphs on hand. Here we ask whether lightly adapting a tabular FM on a small set of source graphs improves its out-of-domain performance on unseen targets (**Q2**).

**Setup.** We fine-tune the underlying TabPFN on nine node classification datasets and evaluate on the remaining 20 datasets. To isolate the effect of fine-tuning over TabPFN, both variants (pretrained vs. fine-tuned) exclude LinearGNNs and use 10 subsampled tables.

We do not compare the finetuned TAG with GFMs trained on additional graphs, since such a comparison conflates two training data sources: TAG benefits from pretraining and additional graph data, whereas GFMs are trained from scratch and only receive graph data. Here, we isolate the within-model effect of adapting a pretrained TFM. By contrast, Section 5.1 reports the cross paradigm comparison, with TAG pretrained on synthetic tabular data and GFMs trained solely on graph data.

**Results.** We observe that fine-tuning yields a modest but consistent average gain of 0.52% across the 20 unseen datasets in Table 2, implying that light adaptation can improve out-of-domain accuracy (**Q2**). Notably, a few datasets (e.g., *wisconsin* and *cornell*) exhibit small declines as they are feature-dominated, with node features contributing far more than topology. As a result, such adaptation toward graph data will induce mild negative transfer.

## 5.3 The importance of depth in feature encoders

**Setup.** To assess the effect of feature encoder depth, we vary the maximum depth $k \in \{1, 2, 3, 4\}$ in a TAG model with 10 subsampled tables and 8 LinearGNNs. For each $k$, we include all feature encoders $\{\phi^{(i)}\}_{i=0}^{k}$ together with the three structure encoders $\{\psi^{(j)}\}_{j=1}^{3}$. LinearGNNs associated with feature encoders deeper than $k$ are omitted.

**Results.** Figure 3 presents the accuracy over 28 tested datasets. On average, TAG with only structure encoders significantly underperforms compared with all other variants, confirming that features are highly informative for the node classification task. Increasing aggregation depth further improves accuracy when $k$ increases from 0 to 2, while the gain plateaus and slightly regresses beyond $k = 2$.

Moreover, we observe two clear dataset-dependent patterns. For the geospatial "airline" graphs (*brazil*, *europe*, *usa*), structural encodings alone suffice: adding $k$-hop features brings little to no benefit, suggesting that node attributes in this family are less informative. In contrast, on heterophilic benchmarks (*wisconsin*, *cornell*, *texas*), informative node features are crucial. Immediate neighbors often have conflicting labels and attributes, so incorporating information from more distant nodes refines each node's embedding, leading to improved performance. Together, these results show that TAG can adapt to both structure-dominated and feature-dominated datasets, emphasizing structural patterns when features contain less signal, while relying more on features and long-range context under heterophilic settings.

## 5.4 THE EFFECT OF THE NUMBER OF SUBSAMPLED TABLES ON THE PERFORMANCE OF TAG

**Setup.** We vary the number of TabPFN models and associated subsampled tables from 1 to 10 to understand their impact on TAG's performance. We report the average accuracy of TAG with 8 LinearGNNs over 28 datasets against the number of subsampled tables (per-dataset results in Section A.2).

**Results.** As expected, TAG's performance increases with the number of subsampled tables, although the rate of improvement gradually flattens. This suggests that additional subsampled tables provide useful complementary information, which our ensemble aggregation mechanism effectively integrates. We hypothesize that increasing redundancy among subsampled tables causes the diminishing gains.

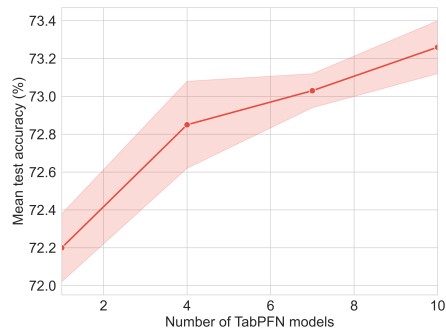

Figure 4: Average accuracy of TAG across 28 datasets vs. the number of subsampled tables.

## 5.5 THE IMPORTANCE OF ENSEMBLE SELECTION AND LINEARGNNS

**Setup.** We initialize TAG with an ensemble comprising 10 TabPFN models and 8 LinearGNNs, and ablate the use of ensemble selection and LinearGNNs to assess their contributions. Specifically, we evaluate two variants: (i) TAG$_{equal}$, which replaces ensemble selection with uniform averaging; and (ii) TAG$_{pure}$, which removes the LinearGNNs, retaining only the tabular foundation models.

Table 3: Accuracy of TAG with and without ensemble selection and LinearGNNs.

| Method | Accuracy |
|---|---|
| TAG | 73.26±0.14 |
| TAG$_{equal}$ | 71.61±0.10 |
| TAG$_{pure}$ | 71.03±0.31 |

**Results.** Table 3 reports mean test set accuracies across the 28 datasets described in Section 5. In all cases, the ablated variants underperform the full TAG model. While LinearGNNs exhibit lower standalone accuracy, their architectural diversity yields error patterns that are less correlated with those of TabPFNs, enabling ensemble selection to construct stronger ensembles. However, the gains from such diversity are fully realized only when ensemble weights are optimized in a principled manner, as naive averaging markedly reduces performance.

## 6 CONCLUSIONS

We introduced TAG, a tabular approach to graph learning that reformulates node classification as a tabular classification problem. TAG represents nodes as rows in a table through a combination of feature and structural encodings, and leverages pretrained tabular foundation models (TFMs), which are applied to subsampled tables and aggregated through ensemble selection. Despite requiring no pretraining on graph data, it outperforms state-of-the-art GFMs and task-specific GNNs by 7%.

We presented initial evidence that fine-tuning TFMs on collections of graph datasets is feasible. A key direction for future work is to scale this process to larger and more diverse collections, comprising both synthetic and real-world graphs, to fully leverage the capacity of TFMs. Another promising avenue is to extend fine-tuning beyond the TFMs themselves to also include the feature and structural encoders, enabling the entire pipeline to adapt jointly to graph-specific structures and representations. Finally, broadening the scope of TAG to tasks such as link prediction and graph classification would further move it towards a general-purpose approach for solving graph tasks.

## ETHICS STATEMENT

This paper presents work aimed at advancing the field of graph machine learning through the development of a tabular approach for graphs, aiming to solve node classification. Potential applications of such models include social network analysis, recommendation systems, fraud detection, and knowledge graph completion. These applications can have significant societal impacts, ranging from improved information retrieval and decision support to risks such as bias amplification, privacy concerns, or the reinforcement of misinformation. However, we do not identify any specific, immediate concerns requiring special attention in the context of this work. We acknowledge and adhere to the ICLR Code of Ethics.

## REPRODUCIBILITY STATEMENT

We provide the full details of our method in Appendix 4 and Section C, along with an anonymous repository containing all scripts and configuration files required to reproduce the results in both the main paper and the appendix. Experiments can be executed on a single NVIDIA RTX PRO 6000 Blackwell GPU with an AMD EPYC 9554 CPU. Unless stated otherwise, all results are reported as the mean accuracy over five random seeds, with the corresponding standard errors.

Baseline results were not reimplemented and are taken directly from Finkelshtein et al. (2025), who evaluated on the same datasets and splits.

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

## A  ADDITIONAL RESULTS

### A.1  THE IMPORTANCE OF STRUCTURE ENCODINGS

Table 4: Average per-dataset accuracy of TAG with and without structure encoders. TAG without structure encoders still uses all feature encoders.

| Dataset | w/o structure encoders | with structure encoders |
|---|---|---|
| actor | $31.24_{\pm0.15}$ | $31.18_{\pm0.16}$ |
| amazon-ratings | $43.63_{\pm0.21}$ | $44.34_{\pm0.29}$ |
| arxiv | $66.67_{\pm0.16}$ | $66.70_{\pm0.18}$ |
| blogcatalog | $79.41_{\pm1.59}$ | $79.77_{\pm0.94}$ |
| brazil | $63.08_{\pm4.16}$ | $73.08_{\pm3.61}$ |
| chameleon | $73.60_{\pm0.69}$ | $72.94_{\pm1.01}$ |
| citeseer | $67.40_{\pm0.77}$ | $68.08_{\pm0.54}$ |
| co-cs | $91.11_{\pm0.11}$ | $90.97_{\pm0.24}$ |
| co-physics | $92.39_{\pm0.28}$ | $92.33_{\pm0.19}$ |
| computers | $84.11_{\pm0.42}$ | $84.33_{\pm0.30}$ |
| cornell | $77.84_{\pm0.90}$ | $75.14_{\pm1.93}$ |
| deezer | $52.33_{\pm1.40}$ | $52.99_{\pm1.48}$ |
| europe | $55.88_{\pm2.17}$ | $55.38_{\pm2.06}$ |
| full-DBLP | $72.84_{\pm0.69}$ | $71.92_{\pm1.30}$ |
| full-cora | $58.85_{\pm0.23}$ | $58.45_{\pm0.22}$ |
| last-fm-asia | $85.09_{\pm0.33}$ | $85.47_{\pm0.26}$ |
| minesweeper | $85.73_{\pm0.10}$ | $85.83_{\pm0.12}$ |
| photo | $90.65_{\pm0.71}$ | $89.63_{\pm0.43}$ |
| pubmed | $78.74_{\pm0.34}$ | $78.96_{\pm0.39}$ |
| questions | $97.11_{\pm0.01}$ | $97.14_{\pm0.01}$ |
| roman-empire | $72.42_{\pm0.20}$ | $74.12_{\pm0.25}$ |
| squirrel | $67.22_{\pm0.40}$ | $65.71_{\pm0.12}$ |
| texas | $81.08_{\pm1.32}$ | $81.08_{\pm1.87}$ |
| tolokers | $82.24_{\pm0.19}$ | $82.82_{\pm0.13}$ |
| usa | $58.38_{\pm1.05}$ | $60.50_{\pm0.71}$ |
| wiki-attr | $70.96_{\pm0.97}$ | $70.09_{\pm0.82}$ |
| wiki-cs | $79.08_{\pm0.44}$ | $79.07_{\pm0.48}$ |
| wisconsin | $82.35_{\pm1.75}$ | $83.14_{\pm1.19}$ |
| Average | $72.91_{\pm0.78}$ | $73.26_{\pm0.76}$ |

**Setup.** To assess the effect of the structure encoder's depth, We evaluate a variant of the TAG model by omitting the structure encoder within 10 subsampled tables and 8 LinearGNNs, and compared it with the standard TAG model. In both cases, we include all feature encoders up to a depth of four.

**Results.** Table 4 shows that incorporating structure encodings consistently improves performance across datasets. In particular, on the "airline" datasets (*brazil*, *europe*, and *usa*), we observe some of the largest gains from adding structure encoders. We attribute these improvements to two complementary benefits of structural encodings. First, while the feature encodings implicitly capture information up to a 4-hop neighborhood around each node, structural encodings make local structural information directly accessible, which may otherwise be difficult to recover from features alone. Second, structural encodings extend the receptive field by enabling access to long-range interactions beyond four hops, which is another likely cause for the observed performance gains.

### A.2  PER-DATASET RESULTS FOR FIGURE 4

In this experiment, we vary the number of subsampled tables and corresponding TabPFN models $B \in \{0, 1, 4, 7, 10\}$. For inference, we ensemble predictions from the $B$ subsampled tables and 8 LinearGNN variants. Per-dataset are reported in Table 5, while Figure 4 visualizes the overall trend together with standard errors.

Table 5: Zero-shot per-dataset accuracy of TAG, with varying TabPFN models $B$.

| Dataset | $B=0$ | $B=1$ | $B=4$ | $B=7$ | $B=10$ |
|---|---|---|---|---|---|
| actor | $31.24_{\pm 0.25}$ | $30.84_{\pm 0.15}$ | $31.16_{\pm 0.23}$ | $31.22_{\pm 0.12}$ | $31.18_{\pm 0.18}$ |
| amazon-ratings | $43.61_{\pm 0.28}$ | $43.90_{\pm 0.15}$ | $44.16_{\pm 0.25}$ | $44.29_{\pm 0.33}$ | $44.34_{\pm 0.32}$ |
| arxiv | $58.43_{\pm 0.05}$ | $58.43_{\pm 0.05}$ | $58.43_{\pm 0.05}$ | $65.72_{\pm 0.27}$ | $66.70_{\pm 0.21}$ |
| blogcatalog | $79.13_{\pm 1.01}$ | $80.25_{\pm 0.99}$ | $79.31_{\pm 0.94}$ | $80.09_{\pm 1.16}$ | $79.77_{\pm 1.06}$ |
| brazil | $53.08_{\pm 3.73}$ | $71.54_{\pm 4.65}$ | $76.92_{\pm 4.55}$ | $73.85_{\pm 1.88}$ | $73.08_{\pm 4.03}$ |
| chameleon | $70.66_{\pm 1.23}$ | $72.72_{\pm 0.85}$ | $72.11_{\pm 1.05}$ | $73.20_{\pm 1.16}$ | $72.94_{\pm 1.13}$ |
| citeseer | $64.68_{\pm 1.03}$ | $65.38_{\pm 1.23}$ | $67.10_{\pm 0.32}$ | $67.12_{\pm 0.75}$ | $68.08_{\pm 0.61}$ |
| co-cs | $90.90_{\pm 0.14}$ | $90.90_{\pm 0.14}$ | $90.97_{\pm 0.19}$ | $91.04_{\pm 0.20}$ | $90.97_{\pm 0.27}$ |
| co-physics | $92.27_{\pm 0.27}$ | $92.14_{\pm 0.26}$ | $92.14_{\pm 0.20}$ | $92.15_{\pm 0.21}$ | $92.33_{\pm 0.22}$ |
| computers | $82.18_{\pm 0.17}$ | $84.03_{\pm 0.68}$ | $84.34_{\pm 0.31}$ | $83.94_{\pm 0.48}$ | $84.33_{\pm 0.33}$ |
| cornell | $75.14_{\pm 1.58}$ | $74.05_{\pm 1.83}$ | $75.68_{\pm 2.09}$ | $76.76_{\pm 1.38}$ | $75.14_{\pm 2.16}$ |
| deezer | $52.08_{\pm 1.28}$ | $51.72_{\pm 1.39}$ | $51.68_{\pm 1.40}$ | $51.67_{\pm 1.61}$ | $52.99_{\pm 1.65}$ |
| europe | $43.25_{\pm 2.72}$ | $55.88_{\pm 1.49}$ | $54.50_{\pm 2.53}$ | $53.50_{\pm 3.12}$ | $55.38_{\pm 2.30}$ |
| full-DBLP | $70.31_{\pm 1.08}$ | $72.43_{\pm 0.79}$ | $73.40_{\pm 1.09}$ | $72.17_{\pm 1.03}$ | $71.92_{\pm 1.46}$ |
| full-cora | $57.18_{\pm 0.25}$ | $57.18_{\pm 0.25}$ | $57.18_{\pm 0.25}$ | $58.44_{\pm 0.17}$ | $58.45_{\pm 0.24}$ |
| last-fm-asia | $84.68_{\pm 0.28}$ | $84.68_{\pm 0.28}$ | $85.12_{\pm 0.29}$ | $85.61_{\pm 0.31}$ | $85.47_{\pm 0.29}$ |
| minesweeper | $82.11_{\pm 0.17}$ | $85.17_{\pm 0.15}$ | $85.74_{\pm 0.10}$ | $85.75_{\pm 0.09}$ | $85.83_{\pm 0.13}$ |
| photo | $90.78_{\pm 0.38}$ | $90.51_{\pm 0.40}$ | $90.80_{\pm 0.62}$ | $90.41_{\pm 0.60}$ | $89.63_{\pm 0.48}$ |
| pubmed | $75.66_{\pm 0.71}$ | $77.68_{\pm 0.67}$ | $76.64_{\pm 1.73}$ | $76.52_{\pm 1.77}$ | $78.96_{\pm 0.43}$ |
| questions | $97.04_{\pm 0.03}$ | $97.03_{\pm 0.03}$ | $97.05_{\pm 0.03}$ | $97.10_{\pm 0.02}$ | $97.14_{\pm 0.01}$ |
| roman-empire | $67.59_{\pm 0.17}$ | $67.59_{\pm 0.17}$ | $73.65_{\pm 0.17}$ | $73.86_{\pm 0.23}$ | $74.12_{\pm 0.28}$ |
| squirrel | $56.64_{\pm 0.47}$ | $64.94_{\pm 0.35}$ | $65.46_{\pm 0.41}$ | $65.57_{\pm 0.14}$ | $65.71_{\pm 0.13}$ |
| texas | $82.70_{\pm 2.78}$ | $82.16_{\pm 2.36}$ | $80.00_{\pm 1.08}$ | $81.08_{\pm 2.09}$ | $81.08_{\pm 2.09}$ |
| tolokers | $79.03_{\pm 0.10}$ | $82.21_{\pm 0.22}$ | $82.82_{\pm 0.16}$ | $82.84_{\pm 0.15}$ | $82.82_{\pm 0.15}$ |
| usa | $50.45_{\pm 2.49}$ | $61.44_{\pm 0.76}$ | $59.78_{\pm 0.92}$ | $60.54_{\pm 0.86}$ | $60.50_{\pm 0.80}$ |
| wiki-attr | $66.18_{\pm 1.88}$ | $66.18_{\pm 1.88}$ | $69.66_{\pm 0.75}$ | $70.26_{\pm 0.98}$ | $70.09_{\pm 0.92}$ |
| wiki-cs | $77.62_{\pm 0.26}$ | $78.96_{\pm 0.60}$ | $79.01_{\pm 0.64}$ | $79.08_{\pm 0.53}$ | $79.07_{\pm 0.54}$ |
| wisconsin | $78.04_{\pm 2.66}$ | $81.57_{\pm 2.29}$ | $85.10_{\pm 0.48}$ | $81.18_{\pm 1.00}$ | $83.14_{\pm 1.33}$ |
| Average | $69.74_{\pm 0.24}$ | $72.20_{\pm 0.18}$ | $72.85_{\pm 0.23}$ | $73.03_{\pm 0.09}$ | $73.26_{\pm 0.14}$ |

## B  DATASET STATISTICS

Dataset statistics for all 28 datasets used in Section 5 are presented in Table 6.

## C  IMPLEMENTATION DETAILS

All reported results are averages across seeds 0, 1, 2, 3, and 4.

**TAG node-level embeddings.** For the GPSE embeddings (Cantürk et al., 2024) (see section 4.1), we use the checkpoint trained on ChEML Gaulton et al. (2012). For RandomWalkPE (Dwivedi et al., 2022) and LaplacianEigenvectorPE (Dwivedi et al., 2023), we pick $k = 20$. Due to computational constraints, we only compute RandomWalkPE for graphs with fewer than or equal to 5000 nodes. TAG filters out the LinearGNN based on RandomWalkPE for datasets with more than 5000 nodes.

**TabPFN.** For all our experiments, we use the newest version of TabPFN at the time of writing. For each TabPFN model, we subsample 2500 random labelled rows and 400 columns. To account for the imbalance in the number of feature-encoding columns vs. the number of structure-encoding columns, 300 of 400 columns are sampled from feature encodings and 100 are sampled from structure encodings.

Since TabPFN natively supports at most ten classes, we adopt the error-correcting output code (ECOC) strategy (Dietterich & Bakiri, 1994) suggested by Hollmann et al. (2025), which splits tasks with more than ten classes into $B$ subtasks of at most ten classes each. Our subsampling strategy is applied independently to each subtask and aggregated outputs form one predictor in the ensemble selection. Given $C$ classes, the ECOC-strategy generally needs at least $\lceil C/9 \rceil$ subproblems to ensure coverage, we do not apply TabPFN models on the dataset with $B < \lceil C/9 \rceil$.

Table 6: Statistics of the 28 node classification datasets.

| Dataset | #Nodes | #Edges | #Feats | #Classes | Train/Val/Test (%) |
|---|---|---|---|---|---|
| actor | 7,600 | 30,019 | 932 | 5 | 48.0/32.0/20.0 |
| amazon-ratings | 24,492 | 186,100 | 300 | 5 | 50.0/25.0/25.0 |
| arxiv | 169,343 | 1,166,243 | 128 | 40 | 53.7/17.6/28.7 |
| blogcatalog | 5,196 | 343,486 | 8,189 | 6 | 2.3/48.8/48.8 |
| brazil | 131 | 1,074 | 131 | 4 | 61.1/19.1/19.8 |
| chameleon | 2,277 | 36,101 | 2,325 | 5 | 48.0/32.0/20.0 |
| citeseer | 3,327 | 9,104 | 3,703 | 6 | 3.6/15.0/30.1 |
| co-cs | 18,333 | 163,788 | 6,805 | 15 | 1.6/49.2/49.2 |
| co-physics | 34,493 | 495,924 | 8,415 | 5 | 0.3/49.9/49.9 |
| computers | 13,752 | 491,722 | 767 | 10 | 1.5/49.3/49.3 |
| cora | 2,708 | 10,556 | 1,433 | 7 | 5.2/18.5/36.9 |
| cornell | 183 | 554 | 1,703 | 5 | 47.5/32.2/20.2 |
| deezer | 28,281 | 185,504 | 128 | 2 | 0.1/49.9/49.9 |
| europe | 399 | 5,995 | 399 | 4 | 20.1/39.8/40.1 |
| full-DBLP | 17,716 | 105,734 | 1,639 | 4 | 0.5/49.8/49.8 |
| full-cora | 19,793 | 126,842 | 8,710 | 70 | 7.1/46.5/46.5 |
| last-fm-asia | 7,624 | 55,612 | 128 | 18 | 4.7/47.6/47.6 |
| minesweeper | 10,000 | 78,804 | 7 | 2 | 50.0/25.0/25.0 |
| photo | 7,650 | 238,162 | 745 | 8 | 2.1/49.0/49.0 |
| pubmed | 19,717 | 88,648 | 500 | 3 | 0.3/ 2.5/ 5.1 |
| questions | 48,921 | 307,080 | 301 | 2 | 50.0/25.0/25.0 |
| roman-empire | 22,662 | 65,854 | 300 | 18 | 50.0/25.0/25.0 |
| squirrel | 5,201 | 217,073 | 2,089 | 5 | 48.0/32.0/20.0 |
| texas | 183 | 558 | 1,703 | 5 | 47.5/31.7/20.2 |
| tolokers | 11,758 | 1,038,000 | 10 | 2 | 50.0/25.0/25.0 |
| usa | 1,190 | 13,599 | 1,190 | 4 | 6.7/46.6/46.6 |
| wiki | 2,405 | 17,981 | 4,973 | 17 | 14.1/42.9/43.0 |
| wiki-cs | 11,701 | 431,206 | 300 | 10 | 5.0/15.1/49.9 |
| wisconsin | 251 | 900 | 1,703 | 5 | 47.8/31.9/20.3 |

**LinearGNNs.** The eight additional LinearGNNs in TAG are each based on one of the eight node-level encodings presented in Section 4.1. We normalize the outputs $\mathbf{l} = (l_c)_{c \in C}$ of the LinearGNNs before ensembling using the following proportional scaling, which maps the smallest logit to 0

$$\mathbf{l}'_c = \mathbf{l}_c - \min_i \mathbf{l}_i + \epsilon \quad \forall c \in [C],$$

$$\mathbf{p} = \frac{\mathbf{l}'}{\sum_i \mathbf{l}'_i}.$$

This ensures that ensemble models with low weights cannot overrule other ensemble members by predicting unbounded logits with high amplitude.

**Ensemble selection.** Our implementation of ensemble selection (Caruana et al., 2004) samples models with replacement, breaks ties (during the greedy selection) randomly and implements early stopping, i.e. it picks the first model configuration in the history that achieved the best possible (accuracy) score on the held-out predictions. We used $k$-fold cross-validation to generate held-out predictions for all labeled nodes $L$. For TabPFN-based models, we use two folds and five folds for LinearGNNs.

**Finetuning.** We use the *cora*, *texas*, *tolokers*, *photo*, *roman-empire*, *usa*, *actor*, *computers*, *europe* datasets for the finetuning experiment in Section 5.2. We finetune TabPFN for 5000 steps using the Adam optimizer with a learning rate of $10^{-6}$ and standard cross-entropy loss. At each step, we sample three datasets and randomly split them into context and query sets according to their original train/test proportions. Before passing the resulting tables to TabPFN, we apply our subsampling strategy described in Section 4.2. For datasets with more than 10 classes, we employ the error-correcting output codes (ECOC) strategy and randomly sample one subproblem, which is then subsampled. These steps ensure that the finetuning tasks resemble the subsampled tables provided to TabPFN

within TAG. We adopt early stopping by holding out each dataset's test set: from the remaining nodes, we select a fixed subset of columns and labeled rows to predict the held-out samples. If the loss on these samples does not improve for $500$ consecutive steps, the run is terminated.

