# OpenReview forum: "Bringing Graphs to the Table: Zero-Shot Node Classification via Tabular Foundation Models"
_ICLR.cc/2026/Conference — ICLR 2026 Conference Withdrawn Submission_

### Official Review · Reviewer_gLng · 2025-10-18

**Soundness:** 1
**Presentation:** 2
**Contribution:** 1
**Rating:** 2
**Confidence:** 5

**Summary:**

The paper proposes applying tabular foundation models such as TabPFN to graph node classification tasks. Specifically, the paper proposes the following approach. First, for each node, various aggregations of its neighborhood features and graph-topology-based structural encondings are computed and concatenated to the original node features, thus enriching these features with additional graph-topology-based information. Then, the obtained table with node features is subsampled to obtain multiple smaller tables that can fit into TabPFN (which cannot work with large tables), and TabPFN predictions for these smaller tables are obtained. Additionally, predictions are obtained for multiple LinearGNN models. In the end, final predictions are obtained as a weighted combination of multiple predictions from TabPFN and LinearGNNs through ensemble selection (the weights are fit on a held-out dataset). The paper evaluates the proposed approach on a collection of 28 node classification datasets.

**Strengths:**

The proposed approach of applying tabular foundation models to graph node classification is interesting and seems promising. Creating multiple smaller subsamples of a single large table and then aggregating prediction obtained for these subsamples is a good way to overcome the limitations of tabular foundation models regarding dataset size.

**Weaknesses:**

The critical weakness of the paper is that its experimental setup is fundamentally flawed. First, the paper evaluates models by computing accuracy on 28 node classification datasets and reporting the mean accuracy across datasets. However, these datasets are very diverse, have different numbers of classes, different class size balance, different node features that are related to node labels in very different ways, and thus different task difficulties and different Bayes error rates. As a consequence, the accuracy values obtained for these datasets are not directly comparable to each other and **cannot** be aggregated by taking their mean. Comparing models by computing the mean performance across several datasets, where on some of these datasets all models achieve performance around 0.3 and on others around 0.8, is absolutely meaningless. It is not surprising that with such a setup the mean accuracy values reported for all baselines do not differ statistically from each other. What is even worse, some of these datasets are binary classification datasets with extreme class imbalance (e.g., in the questions datasets, the majority class constitutes ~97% of nodes), and thus accuracy is not a sound metric for such datasets even taken individually. ROC AUC, as in the original paper proposing the questions dataset [1], or, even better, average precision should be used for such datasets. It can be seen in the reported results that all models achieve very similar accuracy scores on the questions dataset, because accuracy cannot distinguish their performance. A sound approach to such experiments would be to choose appropriate metrics for each dataset and then compare model **ranks** across different datasets.

There are also further issues with the reported experimental results. The results reported for the baselines are often suspiciously weak. For example, on the arxiv dataset, all baselines achieve accuracy values between 0.5 and 0.6. At the same time, on the official OGB leaderboard, even the simplest models GCN and GraphSAGE achieve accuracy values above 0.7 on this dataset. As another example, for multiclass datasets from [1] (roman-empire and amazon-ratings), the accuracy values reported in this paper are significantly lower than those reported in [1].

The dataset selection is also very questionable. The paper reports that their approach achieves particularly strong results on the texas, cornell, and wisconsin datasets, however, these datasets have been criticized in [1] for very small size and extreme class underrepresentation (the texas dataset has a class consisting of a single node). Squirrel and chameleon datasets, also used in this work, have also been criticized in [1] for being buggy. Another set of datasets on which the current work reports strong gains for the proposed method are the brazil, europe, and usa airport datasets. However, these datasets do not have node features (the node feature matrix provided for them in PyTorch Geometric is actually the adjacency matrix). Thus, these datasets cannot be used to evaluate foundation model ability to generalize to different feature spaces, which is one of the key challenges. As can be seen from the provided results, the performance improvements obtained by TAG are disproportionately high on the texas, cornell, wisconsin, brazil, europe, and usa datasets and account for most of the improvement in the average accuracy (although, as discussed above, this metric is meaningless), while for many of the other datasets there is no improvement at all.

As the considered work is entirely empirical, such an unsound experimental setup is a critical flaw and makes the provided results completely untrustworthy. Thus, I cannot recommend acceptance for the paper. New experiments need to be run from scratch in a different setup and with proper baselines to meaningfully evaluate the proposed method. Graph ML suffers from poor benchmarking practices so much that there are even papers being written about this [3], and thus proper experimental setups should be used, especially when tackling such new and important topics as foundation models, and the experimental setup used in the considered paper cannot provide trustworthy results.

There are also other issues. The baseline selection is very limited. Only two GNNs and one GFM are considered. The paper also reports results for GraphAny and states that it is also a GFM, however, I do not think GraphAny can be considered a GFM: it is simply a weighted combination of linear regression models on top of aggregated features, where only the few combination weights are pretrained, while the majority of weights (the linear regression parameters) are fit on each dataset individually. Note that the paper that proposes GraphAny [4] never calls it a GFM.

The paper incorrectly uses the terminology regarding zero-shot prediction. Zero-shot refers to solving a task with access to zero examples (hence, *zero*-shot), similar to how LLMs can answer arbitrary questions without any examples. However, TabPFN, on which the proposed approach is based, uses **all** training examples to make predictions. Hence, it does not do zero-shot predictions, but rather does in-context predictions (because no weight fitting is done on the provided examples). However, as for the proposed approach on the whole, I do not think it can be considered doing in-context predictions, since it does fit some weights on the provided examples - specifically, the weights of LinearGNNs and the weights of different model predictions for ensembling. Thus, the approach is not training-free, as the paper claims.

It is unclear which of the results of the proposed approach are due to the use of TabPFN and which are due to feature engineering (feature and structure encoders). Ablations need to be performed by removing feature engineering (thus limiting TabPFN to only the original node features), and also by providing the baselines with the same combined features (original features plus feature encoders and structure encoders). Also, there have been multiple papers proposing to aggregate neighborhood features or create structural encodings and then apply graph-agnostic (tabular, as the paper calls them) models to these features, e.g. [5-7]. A discussion of this line of research is needed.



[1] A critical look at the evaluation of GNNs under heterophily: Are we really making progress? (ICLR 2023)

[2] https://ogb.stanford.edu/docs/leader_nodeprop/, from Open Graph Benchmark: Datasets for Machine Learning on Graphs (NeurIPS 2020)

[3] Graph Learning Will Lose Relevance Due To Poor Benchmarks (ICML 2025)

[4] Fully-inductive Node Classification on Arbitrary Graphs (ICLR 2025)

[5] Simplifying Graph Convolutional Networks (ICML 2019)

[6] SIGN: Scalable Inception Graph Neural Networks (2020)

[7] Large Scale Learning on Non-Homophilous Graphs: New Benchmarks and Strong Simple Methods (NeurIPS 2021)

**Questions:**

- What exactly is MeanGNN? If it is simply a GNN with mean aggregation, than, contrary to the currently provided reference, it has surely existed way before 2024 (at least GraphSAGE can be mentioned [8]).

- What is the inference time for the proposed approach? I expect it to be significantly slower then that of the baselines due to relatively large size of TabPFN and the need for ensembling.

- Why are only TS-Mean and GraphAny discussed as example of GFMs? Besides GraphAny arguably not being a GFM (see above), there have been many other works on GFMs, some of them mentioned in [9]. Many of these GFMs also tackle node classification. Also, the paper claims that "the GFM pretrained on the largest number of datasets to date is a variant of TS-Mean, which was trained only on nine datasets", which is definitely not correct. For example, GraphFM [10] has been pretrained on 152 datasets (including synthetic ones), and AnyGraph [11] has been pretrained on 18 datasets.

- The paper mentions several times that tabular foundation models can achieve strong generalization on NLP datasets, however, the provided references do not seem to support this - they all discuss tabular datasets, occasionally with additional textual information that is handled with LLMs. Can the authors provide more evidence for this claim?



[8] Inductive Representation Learning on Large Graphs (NeurIPS 2017)

[9] Towards Graph Foundation Models: A Transferability Perspective (2025)

[10] GraphFM: A Scalable Framework for Multi-Graph Pretraining (2024)

[11] AnyGraph: Graph Foundation Model in the Wild (2024)

---

### Official Review · Reviewer_Ghjg · 2025-10-27

**Soundness:** 1
**Presentation:** 3
**Contribution:** 2
**Rating:** 2
**Confidence:** 4

**Summary:**

This work presents TAG, a new approach to solve node classification task by converting graph dataset into tabular format and applying tabular foundation model to make predictions. In particular, this method combines the original node features with smoothed k-hop neighbor features and various graph structural encodings, such as Random Walk positional encodings and Laplacian eigenvectors. After that, it applies TabPFN foundation model on top of the preprocessed node representations to impute missing test labels. The experiments on standard node classification datasets show that the proposed TAG framework outperforms some other alternatives among graph foundation models and standard message passing baselines.

**Strengths:**

1. An interesting idea of converting graph structural information into tabular format and using TabPFN foundation model to solve node property prediction task.
2. Extensive experiments on numerous classic graph datasets for node classification.

**Weaknesses:**

1. The central part of the proposed TAG framework is converting node features and structural information into tabular format and applying tabular foundation model on top of new representations. However, there are no experiments on graph datasets with tabular node features, which should be more relevant domain for the proposed method. Instead, all the considered datasets contain bag-of-words features or textual embeddings that may hardly fit the type of data for which the TabPFN framework has been developed. I would highly recommend to run experiments on the recently proposed GraphLand benchmark [1], which includes graph datasets with tabular node features.
2. Another important part of the proposed TAG framework is aggregating the outputs of TabPFN with the predictions of LinearGNN models trained on top of the original node features and various graph-based encodings. However, one could combine these extra LinearGNN models with any other approach, not only TabPFN. Thus, the motivation for using these ensemble members is not clear, and the proposed combination gives the impression of a well-chosen approach for a Kaggle competition, rather than a carefully studied and necessary component. To explain my point, consider using properly tuned and potentially deeper GNNs instead of the considered LinearGNNs: it is natural to expect that such an ensemble of TabPFN and GNN models can achieve even better results. However, this approach can quickly become AutoML, where we simply optimize the metric by any means possible, without investigating how and why the ensemble members complement each other.
3. No ablation study on the impact of each structural encoding used in the TAG framework. It is not clear how much each of them affects the final performance, and whether it is possible to supply standard message passing baseline with such structural encodings and achieve comparable results. Moreover, according to Table 4, completely rejecting structural encodings in most cases does not lead to notable performance degradation, and sometimes even improves the metric, which questions the necessity of using them in the proposed framework.
4. No details about hyperparameter tuning and design of neural architecture for message passing baselines. It has been shown [2] that proper hyperparameter selection for graph neural networks supplied with skip-connections and normalizations can significantly improve their performance. Thus, conducting thorough hyperparameter search and using simple architectural improvements is necessary to obtain fair and reliable results.

[1] GraphLand: Evaluating Graph Machine Learning Models on Diverse Industrial Data, NeurIPS 2025

[2] Classic GNNs are Strong Baselines: Reassessing GNNs for Node Classification, NeurIPS 2024

**Questions:**

1. Why do the authors obtain such high standard deviation for standard message passing baselines? So far, the superior average performance of TAG framework seems to be the consequence of particular dramatic drops in predictive performance of baseline models rather than consistent improvement of the proposed method. Moreover, it is hard to understand to what extent the presented results correspond to the level that the community has managed to achieve. For instance, comparing the results in Table 1 with the aforementioned study [2] on strong GNN baselines: the reviewed paper reports ~44 points of accuracy on `amazon-ratings` versus ~55 points that can be achieved by properly tuned baselines; ~74 points on `roman-empire` versus ~91 points in baselines; ~67 points on `ogbn-arxiv` versus ~73 points in baselines. The comparison on other datasets is complicated because of the different metrics used for evaluation — accuracy in this work and AUROC in the study on GNN baselines.
2. Why do the authors average performance on different datasets with different data splits and different number of classes? The accuracy metric in this case can vary significantly and may not allow to compare it across different settings. Thus, such an approach to present average predictive performance can be misleading, especially given such high standard deviation in the reported metrics. I would recommend to compute average ranks instead.
3. Why do the authors train TAG on some number of graph datasets unrelated to the prediction task of interest and do not consider finetuning on the train part of the dataset that is exactly used for evaluation? In terms of the final predictive performance, such an approach should be more productive, since the model does not learn to solve some unrelated tasks, but is focused precisely on the problem it needs to solve now. This might be supported by the fact that, according to Table 2, the difference between pretrained and finetuned model on most datasets appears to be statistically insignificant, and sometimes the performance of the finetuned model is even worse.
4. I think that the usage of term *zero-shot* in this work is misleading. To the best of my knowledge, by *zero-shot* one usually means a setting when the model gets no labeled examples and must solve the given task without input-output pairs. However, the presented TAG framework is based on the TabPFN foundation model that can produce reasonable predictions *exactly* because it is provided with a certain number of labeled train examples, as it exploits in-context learning mechanism and does not require explicit training to solve the given task. Thus, I would highly recommend to avoid using the term *zero-shot* in this work and instead refer to in-context learning.

---

### Official Review · Reviewer_bVz5 · 2025-10-29

**Soundness:** 2
**Presentation:** 2
**Contribution:** 2
**Rating:** 4
**Confidence:** 4

**Summary:**

This paper proposes a novel approach to leverage tabular foundation models (TFMs) for graph node classification tasks. The key idea is to transform graph data into a tabular format by constructing node-level features, thereby representing each node as a tabular data point. This is achieved through the design of node feature encoders and graph structural encoders, which collectively convert the original graph into a large table suitable for TFM processing. To address the inherent limitations of TabPFN with respect to sample and feature size, the authors further introduce subsampling and ensemble strategies. Extensive experiments are conducted on 28 node classification datasets, comparing the proposed method with two end-to-end graph baselines and two pretrained graph foundation models. The proposed approach achieves the best performance among these four methods. Additional ablation studies are provided to demonstrate the effectiveness of the proposed components.

**Strengths:**

- The paper is well-motivated, clearly written, and the problem formulation is concise.
- The experimental evaluation is comprehensive, conducted across 28 datasets, with clearly presented results demonstrating strong performance.
- The inclusion of thorough ablation and sensitivity analyses validates the effectiveness of the proposed components (e.g., TFM fine-tuning, ensemble sizing, and the LinearGNN component).

**Weaknesses:**

- The baseline set is too narrow to validate the paper's strong claims that feature engineering plus TFMs is "sufficient." It omits critical GNNs (GCN, GraphSAGE), advanced GNNs (BWGNN, FAGCN), self-supervised graph models, and other relevant Graph Foundation Models (e.g., AnyGraph, OpenGraph).
- The entire evaluation hinges on TabPFN. The generalizability of the graph-to-table transformation is questionable without testing it on other TFMs like TabPFNv2, TabICL, or MITRA.
- The paper does not adequately analyze *which* engineered features (structural vs. node attributes) are most important. This is a critical omission for a method based entirely on feature engineering.
- The ablation studies offer few surprising insights; many findings are expected (e.g., fine-tuning improves performance, ensembling helps). The role of the LinearGNN component feels detached from the core TFM contribution.

**Questions:**

- Can the authors include a broader set of baselines, including common GNNs (GCN, GraphSAGE), self-supervised graph models, and other recent Graph Foundation Models (AnyGraph, OpenGraph), to fully substantiate the paper's claims?
- How well does the proposed feature engineering generalize to other tabular foundation models beyond TabPFN, such as TabPFNv2 or MITRA?
- Can the authors provide a detailed feature importance analysis to disentangle the contributions of structural encoders versus node feature encoders? Are structural features always necessary, or do node attributes suffice in some cases?
- Have the authors considered using the graph adjacency matrix to inform the cross-sample attention mechanism in TabPFN, as an alternative to explicit structural feature engineering?

---

### Official Review · Reviewer_q78s · 2025-10-30

**Soundness:** 2
**Presentation:** 3
**Contribution:** 2
**Rating:** 2
**Confidence:** 4

**Summary:**

This paper focuses on the design of graph foundation models. Specifically, the authors transform the input graph into tabular-style data via feature and structural encoders. In this way, the tabular foundation model-based methods could be used for learning node representations. The experimental results showcase the effectiveness of the proposed method for downstream tasks.

**Strengths:**

1.This paper is easy to follow.

2.Transforming the input graph into other style data seems to be interesting.

3.The experiments are extensive.

**Weaknesses:**

1.The research gap is not clear.

2.The overall technique contribution is somewhat limited.

3.The experimental results lack deep discussions.

**Questions:**

1.I think the research gap is not clear. The authors argue that “existing GFMs are often trained on datasets that not fully reflect real-world graphs”. However, the authors do not provide enough supports for transforming graph data into tabular-style data can solve the above issue.

2.Encoder module and learning module are tow core modules of TAG. As the authors mentioned, these modules adopt many popular techniques which could be regarded as a combination of existing strategies. For instance, the node encoding module should be the most important part. However, TAG still leverages common techniques, such as multi-hop neighborhood aggregation and structural encoding for this part which significantly weakens the technique contributions.

3.The discussions about the experimental results are not deep. Moreover, the authors claim that “Table 4 shows that incorporating structure encodings consistently improves performance across datasets.” which is against the results shown in Table 4.

---

### Author Response · Authors · 2025-11-20

We thank the reviewers for the time and effort invested in reviewing our paper. We greatly appreciate the constructive feedback and will carefully incorporate it as we work on the next iteration of our work.

We would therefore like to formally withdraw our submission at this time.

---

### Note · Authors · 2025-11-20

I have read and agree with the venue's withdrawal policy on behalf of myself and my co-authors.